# Lightweight and Energy-Efficient Deep Learning Accelerator for Real-Time Object Detection on Edge Devices

**DOI:** 10.3390/s23031185

**Published:** 2023-01-20

**Authors:** Kyungho Kim, Sung-Joon Jang, Jonghee Park, Eunchong Lee, Sang-Seol Lee

**Affiliations:** Intelligent Image Processing Research Center, Korea Electronics Technology Institute, Seongnam-si 13488, Republic of Korea

**Keywords:** tiny machine learning (TinyML), internet of things (IoT), deep learning, hardware accelerator, edge devices, object detection, field-programmable gate arrays (FPGA)

## Abstract

Tiny machine learning (TinyML) has become an emerging field according to the rapid growth in the area of the internet of things (IoT). However, most deep learning algorithms are too complex, require a lot of memory to store data, and consume an enormous amount of energy for calculation/data movement; therefore, the algorithms are not suitable for IoT devices such as various sensors and imaging systems. Furthermore, typical hardware accelerators cannot be embedded in these resource-constrained edge devices, and they are difficult to drive real-time inference processing as well. To perform the real-time processing on these battery-operated devices, deep learning models should be compact and hardware-optimized, and hardware accelerator designs also have to be lightweight and consume extremely low energy. Therefore, we present an optimized network model through model simplification and compression for the hardware to be implemented, and propose a hardware architecture for a lightweight and energy-efficient deep learning accelerator. The experimental results demonstrate that our optimized model successfully performs object detection, and the proposed hardware design achieves 1.25× and 4.27× smaller logic and BRAM size, respectively, and its energy consumption is approximately 10.37× lower than previous similar works with 43.95 fps as a real-time process under an operating frequency of 100 MHz on a Xilinx ZC702 FPGA.

## 1. Introduction

Deep learning has been popular because of the availability of computing power and the development of big data [1], and various reviews and discussions on deep learning have been extensively conducted in recent years [2,3,4,5,6]. It has been widely applied in many fields such as image recognition [7], object detection [8], autonomous driving [9,10], and robotics [11]. Moreover, deep learning networks have been shown to be successful for these fields [12], and nowadays it has become important even in the field of IoT with the rapid development of IoT devices and network infrastructure [13]. Accordingly, deep learning operation in real time with low energy on these resource-constrained edge devices has emerged as essential work in the era of IoT [14].

However, deep learning models are generally too complex, and they also require considerable amounts of data and their computation [15]. Model complexity in deep learning is a fundamental issue in terms of model framework, model size, optimization process, and data complexity. Most deep learning models have a complex model framework, such as a convolutional neural network (CNN), and their model size is so huge owing to numerous parameters, layers, and filters. In addition, the configuration, such as layer width and filter size, also affects model size. As a result, running a deep learning model requires so much memory to store those numerous parameters and a tremendous amount of intermediate data. Furthermore, high energy consumption is inevitable for computing their calculation and moving so much data from/to memory. Therefore, it is very challenging to implement the hardware to accelerate these deep learning models on the battery-operated IoT devices.

Thus, TinyML is an important and emerging area for operating machine learning applications on small embedded IoT devices, and hence it has been actively researched recently [16,17,18,19,20]. It aims at designing and developing algorithms and hardware capable of performing inferences on resource-constrained devices at extremely low energy. Accordingly, it takes into account the characteristics of hardware to be operated and tries to optimize the model for the hardware and reduce computational load and memory demand by deploying approximation and compression, like pruning. Moreover, the hardware design has to be implemented as lightweight to be embedded in low-cost resource-constrained devices and energy-efficient so that the deep learning model works smoothly on the battery-operated devices, and low-latency so as to run in real time on IoT edge devices, considering the fast sensory data streams.

In this paper, we present the optimized network model for hardware to be implemented. The proposed optimized model is based on SqueezeNet [21], which is a mobile-oriented network. We perform model reduction and parameter simplification on the backbone model network through model simplification, and integer quantization is adopted for activation and parameters through model compression. Furthermore, a lightweight and energy-efficient hardware architecture is proposed, and an implemented design is able to perform parallel processing between layers and channels deploying a 3D tensor-like processing element (PE) structure. It results in low latency and reduction in energy consumption. Besides, a small amount of on-chip memory is required owing to the proposed on-chip memory management strategy, which makes the design lightweight and low-power. As a result, the experimental results demonstrate that our optimized model successfully performs object detection, and the proposed hardware design achieves 1.25× and 4.27× smaller logic and BRAM size, respectively, and its energy consumption is approximately 10.68× lower than previous related works with 43.95 fps as a real-time process under an operating frequency of 100 MHz on a Xilinx ZC702 FPGA.

The rest of this paper is structured as follows: Section 2 provides some background of lightweight deep learning techniques with our backbone model and its related works. Section 3 presents the proposed model optimization through model simplification and compression for hardware to be implemented. In Section 4, the proposed hardware architecture is presented for the hardware design to be lightweight and energy-efficient. Experimental results for the performance of the proposed model and hardware architecture are shown in Section 5. Lastly, Section 6 discusses the conclusion.

## 2. Background

Operating deep learning models on edge devices is quite challenging because of their limited resources and computational capabilities. Thus, it is important to make use of lightweight deep learning models that are suitable for execution on resource-constrained devices. Moreover, it is crucial to design lightweight and energy-efficient hardware owing to the battery limitations of low-cost devices. Consequently, hardware/software (HW/SW) co-optimization is critical to deploying deep learning models on these devices, and considerable related research has been aggressively conducted as well [22,23,24].

Lightweight deep learning techniques are able to be classified into two categories: lightweight deep learning algorithms and transforming existing models into compact/small ones [25,26]. SqueezeNet [21], as a representative edge-device-oriented deep learning model, is a lightweight deep learning algorithm. This category makes the structure of the network model lightweight to reduce computational complexity and the number of parameters by utilizing a residual block or bottleneck block. In addition, converting existing models into compact ones is achieved by knowledge distillation or model compression, such as pruning, quantization/binarization, and the weight-sharing method. These techniques compress the model size and its computation by eliminating redundant parameters, sharing common values, and reducing data bits. Many studies [27,28,29,30] in this field have progressed deploying deep learning models on the edge devices.

The design and implementation of hardware for algorithms, as well as the utilizing of lightweight algorithms, are also crucial for the practical operation on the resource-constrained IoT devices. It should work with low latency, power, and energy with lightweight designs on these devices. Related studies [31,32,33] on efficient hardware design have been actively conducted to determine the feasibility of running deep learning models on edge hardware. Furthermore, hardware design, even in low-cost devices, should be optimized and customized with optimal architectures. On-chip or external memory and logic are unnecessarily able to be consumed with a general PE configuration that does not consider the characteristics of the deep learning model for deployment.

### 2.1. SqueezeNet

SqueezeNet [21], shown in Figure 1, is a representative lightweight deep learning model in terms of model size and the number of parameters for hardware with limited resources, particularly memory, and computational capabilities. It preserves its accuracy with fewer parameters than AlexNet [34]. 

A micro-architectural view is shown in Figure 1a. The base unit structure, called the fire module, comprises a squeeze layer with a 1 × 1 convolution filter and an expand layer that has a mix of 1 × 1 and 3 × 3 convolution filters. A ReLU process is performed on the output of the squeeze layer, which feeds into the expand layer, and the final output of the fire module comes from a ReLU operation on the output of the expand layer. Figure 1b shows the macro-architectural view. SqueezeNet consists of a series of fire modules and several maxpool functions between the fire modules.

This model has 9× fewer parameters when replacing the generally used 3 × 3 filters with 1 × 1 filters in the squeeze layer. In addition, the base unit fire module makes it possible to reduce the number of channels by deploying a 1 × 1 convolution filter instead of a 3 × 3 convolution filter, and the number of channels is expanded again by deploying 3 × 3 convolution filters in the expand layer. Moreover, the effect of compressed image information can be obtained through sparse down-sampling between a series of fire modules, which reduces the range of the region of interest (RoI) at one glance and leads to higher classification accuracy.

### 2.2. Related Works

Many studies on accelerating SqueezeNet on FPGA have been introduced in the literature [35,36,37,38,39]. In [35], the authors attempted to enhance the performance of the hardware processing convolution operation through pipelining and loop unrolling and flattening. However, it did not affect the performance owing to the bandwidth bound. In addition, they fused convolution and maxpool operations as layer dimensions, but it had a trivial impact on optimization because their implementation result for resource utilization was too large, and the power dissipation was also too large to deploy the model on edge devices. 

Another design and implementation of SqueezeNet, layer-based structured design, was introduced in [36]. The purpose of this design is scalability in constructing CNNs, and it allows the flexible and scalable deployment of the entire CNN. Owing to these characteristics, a large amount of resource utilization was exploited, although a (8–16)-bit fixed quantization strategy was adopted. Moreover, the power consumption was too high, making the design difficult to be embedded and operated on resource-constrained devices.

In [37], they implemented their accelerator with eight multiply-accumulate (MAC) −16 units, which performed 16 MACs in every clock cycle of its operation. In addition, they employed a quantization strategy for parameters such as 8 bits weights, bias, and 16 bits feature maps to reduce their accelerators. In addition, they used various buffers for the parameter and input feature map as an input feature map tile buffer (ITB) and input feature map tile buffer window (ITBW), respectively, avoiding redundant memory accesses that introduce additional power consumption. As a result of their efforts, the power consumption was sufficient to operate on embedded devices, but the memory size was very large, owing to the strategy of utilizing many buffers. Furthermore, the execution time was too long, and therefore an enormous amount of energy was consumed, which made it impracticable to deploy their accelerator on battery-operated edge devices in real time.

A high-speed hardware accelerator was implemented in [38]. The researchers used a ping-pong memory strategy and deployed several first-in, first-outs (FIFOs) in their design to solve the memory bottleneck issue. By preparing a set of twin memories, data from all the series of fire modules can be processed using this ping-pong memory and alternating between a set of twin memories. Additionally, several intermediate FIFOs hold the output data as some pixels of the squeeze layer and pass them to the expand layer when 3 × 3 window data have been filled. Besides, the authors made use of hardware resources with different configurations in the squeeze and expand layers to speed up layer processing. Consequently, they achieved quite low latency of their hardware accelerator, but their resource utilization on logic and memory was quite high because of the twin memory and FIFOs strategy. Accordingly, it was not suitable for embedding on resource-constrained edge devices. Furthermore, the power consumption was also high, making it difficult to operate the deep learning model on low-power devices.

To deploy deep learning models for applications such as object detection and image classification on IoT devices, the hardware accelerator embedded on these devices should be lightweight, low-latency, low-power, and low-energy with real-time processing, considering the characteristics of edge devices. To achieve these factors, HW/SW co-optimization is necessarily required in algorithm and hardware.

## 3. Proposed Model Optimization

To obtain well-optimized hardware suitable for deployment on edge devices, transforming deep learning models into compact or small ones is necessary, as well as making use of lightweight deep learning algorithms. Data quantization is a representative method of model compression for operating a model on hardware with limited resources. Additionally, model simplification for optimizing and customizing the hardware to be implemented is crucial for hardware to be more lightweight and energy-efficient. Therefore, we present a model simplification by reducing the model and simplifying the parameter configuration and model compression with data quantization.

### 3.1. Model Simplification

SqueezeNet, the mobile friendly deep learning model used as our backbone, is a lightweight deep learning algorithm, but it is inevitable that additional optimizations such as model reduction and parameter configuration simplification must be performed, making it executable on low-cost edge devices. In general, the depth of the deep neural network (DNN) is a fundamental issue in terms of the accuracy of the model, the complexity of computation, and runtime. For instance, the deep network shown in Figure 1b involves complex computations with many parameters and intermediate data with high latency and energy consumption, which is not appropriate for low-cost resource-constrained devices. There is a tradeoff between these performances, however; deeper networks or increasing the depth of networks is not always good [40]. Inspired by this, we have conducted model reduction for the model to be lightweight by reducing the depth of the model and involving the intermediate maxpool functions in convolutions with less computation complexity and low latency. Moreover, simplification of the parameter configuration was also performed for the hardware-oriented structure. This is illustrated in Figure 2.

An overview of the simplified model is shown in Figure 2a. It consists of 14 convolutions, 1 maxpool function, and 4 concatenation procedures with several intermediate ReLU processes. One convolution with a ReLU as a squeeze layer and two convolutions with two ReLUs as an expand layer and a final concatenation procedure constitute a fire module; hence, the reduced model has four fire modules in contrast to the eight fire modules of the backbone model in Figure 1b. However, we achieved a successful object detection performance with this reduced model network, as shown in Section 5. In addition, several convolutions have maxpool functions within by adopting stride 2 in the middle of the convolution operation, as shown in Figure 2b. The convolution operations in conv6, conv7, conv9, and con10 in Figure 2b are processed with stride 2 instead of additional maxpool function operations after each convolution process, as shown in Figure 1b. This results in low latency, less computation, and even low energy consumption by involving maxpool functions in the convolution process. In addition, the configuration of the channel number of the filter in Figure 2b indicates a multiple of four, except for the initial primary input, which is three: R, G, and B. This is a constraint for the model to be hardware-oriented, leading to lightweight and energy-efficient hardware with good optimization for resource-constrained devices. The significance of the channel number of the filter to be a multiple of four is described in Section 4.5.1 in detail. 

### 3.2. Model Compression 

Data quantization, a representative model compression technique, is necessarily required to compress a model on resource-limited, low-latency, and low-energy devices owing to the latter’s constraints on compactness and battery capability. Corresponding to the limited memory, computation, and power of these devices, data moving from/to memory and to be processed should be quantized from floating point numbers for small size and low computation and power. The quantization procedure is described in Equation (1):(1)Dataq=Dataf∗SD=∑ActfWf+Bf∗SD=∑Actq/SAWq/SW+Bq/SB∗SD=SD/SASW∗∑Actq∗Wq+SASWSBBqording to the comment.cronyms.ai and about abbreviations and acronyms.

This is a convolution operation utilizing symmetric quantization. Data comprises activation, weight, and bias components denoted as Act, W, and B, respectively. S indicates a scale factor, and D, f, and q denote data, floating point, and quantization, respectively. Floating point elements are able to be classified into quantized elements and their respective scale factors. For instance, $Act_{f}$, floating point activation, is classified into $Act_{q}$, quantized activation, and $S_{A}$, scale factor of activation. Additional calculations using the zero point in this equation are required for asymmetric quantization. The asymmetric quantization typically has a higher resolution than the symmetric quantization. Thus, the weight parameters were quantized by the symmetric quantization, and the asymmetric technique was adopted for activation quantization.

Data quantization for the convolution operation is shown in Figure 3. It describes the quantized bit of each component with configurations. The weight parameter was quantized to integer 8 bits, and bias, multi-scale, and shift-scale factors were quantized to integer 32, 12, and 8 bits, respectively, in the order of BQ, that is, bias and quantization, as shown in Figure 3. In addition, the activation component was also quantized as 8 bits. These quantized parameters are able to be obtained offline so that employing these quantized elements is sufficient to operate inference process for deploying the deep learning model on devices. Consequently, memory size and computational complexity can be reduced, leading to the feasibility of model deployment on low-cost edge devices. 

## 4. Proposed Hardware Architecture

The design and implementation of hardware are very important for deep learning models to be deployed on resource-constrained edge devices, even if the model has a lightweight algorithm that adopts model compression and simplification. In other words, it is difficult to deploy a lightweight model on such devices when the implemented hardware is bulky with redundant logic and memory requirements and does not have a well-optimized architecture, which leads to the huge size of the hardware resource, high latency, power, and energy consumption. Therefore, we propose a hardware architecture with an optimal design for the presented lightweight deep learning model by utilizing parallel processing between layers and channels through a 3D tensor-like PE structure, and a memory-efficient on-chip memory management strategy. 

An overview of the proposed hardware architecture is shown in Figure 4. We performed a customized direct memory access (DMA) design that interacts with an external memory through advanced extensible interface 4 (AXI4) protocol. The inform layer unit controls the order of the layer, and the ifmap (input feature map) driver unit fetches the input feature map data from an external memory through DMA and conducts read/write operations from/to on-chip memories for the ifmap, feeding into an arithmetic core unit. In addition, parameter data such as weights, bias, multi-scale factors, and shift-scale factors are fetched by the parameter driver unit, which reads and writes the fetched parameter data from/to on-chip memories for the parameters. The arithmetic core unit performs convolution and maxpool operations with ifmap data from the ifmap driver unit and parameter data from the parameter driver unit, and the result of the arithmetic core unit is fed into the ofmap (output feature map) driver unit. Finally, the ofmap driver unit performs read/write operations of data output from the arithmetic core unit from/to the on-chip memories for ofmap. The detailed operation of each unit is as follows.

### 4.1. Inform Layer Unit

The layer order is controlled by the inform layer unit, considering the status signal from the ifmap driver unit. The status signal includes information if the loading ifmap data in the current layer has been finished, indicating the availability of the next layer. As a result, the ifmap driver unit and parameter driver unit can be synchronized using the attribute signal as the output signal of the inform layer unit.

### 4.2. Ifmap Driver Unit

#### 4.2.1. Ifmap Read Control (IRC) Unit 

The ifmap read control (IRC) unit fetches ifmap data line by line from an external memory through DMA with a ready signal. The IRC unit passes the loaded ifmap line data to the matrix generation (MG) unit, as described in Section 4.2.2, considering the status signal from the MG unit. Accordingly, this unit enables the MG unit to transform ifmap data to matrix-type.

#### 4.2.2. Matrix Generation (MG) Unit

The MG unit transforms ifmap data to matrix-type for parallel processing between layers, leading to a reduction in latency. In other words, this unit makes it possible to process 3 × 3 and 1 × 1 convolutions in an expand layer simultaneously. The matrix generation process in the MG unit is illustrated in Figure 5. The ifmap stream comes to the MG unit line by line from top to bottom of the input feature map in the order of blue, grey, orange, and green. The first line of the ifmap, the blue one in Figure 5, is stored in on-chip memory 0, while on-chip memories 1 and 2 are idle because they are waiting for the next two lines of ifmap, grey and orange in Figure 5. When the second line, the grey one in Figure 5, streams, the first 3 × 3 matrix output, the dark blue one in Figure 5, comes out with the read state of on-chip memory 0 for the blue ifmap and the read/write state of on-chip memory 1 for the grey ifmap in the case of zero padding on the top line of the 3 × 3 matrix, which finally consists of zeros on top and blue ifmap data in the middle, and grey ifmap data on the bottom. The second 3 × 3 matrix output, the dark grey one in Figure 5, is generated when the third ifmap, the orange one in Figure 5, stream comes with read/write operations on on-chip memory 2. This second 3 × 3 matrix consists of blue ifmap data on top and grey ifmap data in the middle, and orange ifmap on the bottom. Finally, the third 3 × 3 matrix output, the dark brown one in Figure 5, comes out with overwrite and read operations of the fourth ifmap line, the green one in Figure 5, on on-chip memory 0. In this third 3 × 3 matrix, grey, orange, and green ifmap data are located on top, middle, and bottom, respectively. Accordingly, the generated 3 × 3 ifmap matrix includes ifmap data for both 3 × 3 and 1 × 1 convolutions at the center of the matrix.

By this parallel processing between layers with 3 × 3 matrix generation, external memory access can be reduced by half compared to the process in the order of 1 × 1 and 3 × 3 convolutions in an expand layer because the external memory access needs double for feeding the inputs to the 1 × 1 and 3 × 3 convolutions, respectively, at different times. However, this external memory access is the representative issue degrading the performance of the system because it relatively requires so much time and power. Thus, this 3 × 3 ifmap matrix generation enables the embedded hardware to operate in real time with low latency and low energy. In addition, it can also reduce the space for external memory and on-chip memory. If the two layers in an expand layer are processed in sequence, the ifmap as an input of the expand layer has to be stored in memory by the end of the 1 × 1 convolution process because the 3 × 3 convolution should take the same input, resulting in a redundant use of reusable memory. Therefore, redundant memory use or occupancy can be eliminated by the MG unit.

### 4.3. Parameter Driver Unit

#### 4.3.1. Parameter Read Control (PRC) Unit 

The parameter read control (PRC) unit fetches parameter data, such as weights, bias, mult-scale, and shift-scale, from an external memory through DMA with a ready signal. The PRC unit passes the loaded parameter data to the parameter set (PS) unit, as described in Section 4.3.2, considering the status signal from the PS unit. Accordingly, this unit helps the PS unit set the parameter data on time.

#### 4.3.2. Parameter Set (PS) Unit

Parameter data are set in advance as registers for each layer by the PS unit, fed into an arithmetic core unit. The PS unit is synchronized with the MG unit for layer order by the inform layer unit, and the data of each parameter to the arithmetic core unit are also fed at the same time as the ifmap matrix data from the MG unit to the arithmetic core unit. In addition, this PS unit performs a read operation to load parameter data from an external memory through the DMA and a write operation on its own on-chip memories, leading to less external memory access.

### 4.4. Arithmetic Core Unit

#### 4.4.1. Conv Unit

All convolution operations are conducted in a conv unit. The synchronized ifmap matrix data from the MG unit and quantized parameter data from the PS unit are fed into 3D tensor-like PEs, as shown in Figure 6, indicating a 3 × 3 convolution case. It has channel (C), height (H), and width (W) components, as CxHxW type, and comprises 3 × 3 PEs with four numbers of channels in detail, as mentioned in Section 3.1. These 3D tensor-like PEs enable parallel processing between channels, leading to low-latency hardware, and memory-efficient architectures, as discussed in Section 4.5.1. Moreover, the detailed operation on convolution with the quantized data, inside one PE in 3D tensor-like PEs, is shown in Figure 7. Unsigned 8-bit input activation, signed 8-bit weight parameter, signed 32-bit bias parameter, unsigned 12-bit multi-scale factor, and unsigned 8-bit shift-scale factor enter the PE, having multipliers, adders, accumulator, and shifter with a clamping operation.

#### 4.4.2. Pool Unit

The pool unit operates as a maxpool function in the sequence of layers. It takes 3 × 3 ifmap matrix data and performs maxpooling among 9 ifmap data within a 3 × 3 matrix. The output comes into the ofmap write (OW) unit, as described in Section 4.5.1, in the ofmap driver unit in the maxpool layer.

### 4.5. Ofmap Driver Unit

#### 4.5.1. Ofmap Write (OW) Unit

The output feature map data from the arithmetic core unit are fed into the OW unit, storing this output feature map on its own on-chip memory, as shown in Figure 8. Figure 8a,b show the on-chip memory status of the OW unit at conv1 and maxpool layer, respectively. The physical on-chip memory has a 256 × 64 configuration as height (address) × width (bit). As for C1_L0_0, C1 and L0 stand for the first channel and line of output feature map, respectively, and the last number 0 indicates the first output feature map data. In other words, C1_L0_0 indicates the first output feature map data in the first line of the output feature map at the first channel, which is generated one by one, in the order of channel number from C1 to C16 and the order of output feature map width direction from 0 to 127 within a channel, by the arithmetic core unit in the convolution operation, as shown in Figure 8a. Similarly, four output feature map data, C1_L0_0, C2_L0_0, C3_L0_0, and C4_L0_0, are generated at the same time in a maxpool operation as shown in Figure 8b. By gathering the next output feature map data, the data packet {C1_L0_0, C1_L0_1} can be written in the on-chip memory at the conv1 layer and {C1_L0_0, C1_L0_1, C2_L0_0, C2_L0_1, C3_L0_0, C3_L0_1, C4_L0_0, C4_L0_1} can be stored in the on-chip memory at the maxpool layer.

As shown in Figure 8, the entire data in all channels for one line of the feature map are stored in the on-chip memory and split into four channel sections. This indicates that the arithmetic core unit is able to generate the output feature map data without any bottleneck by fetching the entire data of the input feature map in one line across all channels in the next layer because the entire data across all channels are needed to generate the output feature map data. If the entire data are not able to be provided continuously across all channels, several on-chip memories are additionally required to store the partial sum result. This sequential provision across all channels can be obtained by fetching the data in the order of on-chip memory address owing to the split four-channel section. If the channel section is split into more pieces, for example, eight channel sections, lots of data can be loaded at the same time, but that reduces the flexibility and performance of an algorithm. Similarly, if the channel section is split into fewer pieces, for example, two channel sections, the flexibility of an algorithm can be improved, but some data can be fetched at the same time, leading to the low latency of the hardware. Therefore, the significance of the channel number of the filter to be a multiple of four is on HW/SW co-optimization. 

As a result, this configuration of data storage on the on-chip memory of the output feature map is maintained in an external memory so that the input feature map with the same data configuration can be loaded on the on-chip memory of the input feature map. The OUT (IN) on top of the on-chip memory in Figure 8 indicates that the configuration of the output data stored in memory in the current layer is maintained at the input feature map fetch in the next layer. This on-chip memory management strategy enables incremental external memory access, thereby reducing the latency and power consumption caused by frequent and irregular external memory access. As a result, low latency and low energy consumption can be achieved with a small amount of on-chip memory, enabling the implemented hardware to be more lightweight and suitable for the resource-constrained and battery-operated edge devices.

Figure 9 shows the overall process flow of the implemented hardware, which is related to the data read/write through on-chip memories as described in Figure 8. As shown in Figure 9a, the configuration is 16 × 96 × 128 as channel × height × width, and the first line of the output feature map in the first channel is written first in the order of output feature map width direction from 0 to 127 within a channel in the on-chip memory. Next, the first line of the output feature map in the second channel is written in the same order as the first channel in the on-chip memory. As a result, the entire data in the first line of the output feature map across all channels are written in 16 iterations in the conv1 layer. Moreover, this output feature map data is the same as the input feature map data at the same time in Figure 9b, which indicates the ifmap read operation at maxpool layer as the next layer of conv1. Therefore, the input feature map data across four channels can be loaded incrementally without irregular memory access at the maxpool layer, and the input feature map across all 16 channels is able to be fetched in four iterations, as shown in Figure 9b. In addition, these data as matrix-type are fed into the 3D tensor-like PEs in the arithmetic core unit.

#### 4.5.2. Ofmap Read/Write Control (ORWC) Unit

Output feature map data written in their on-chip memories are loaded by the ofmap read/write control (ORWC) unit, which writes the loaded data in an external memory through DMA. A pair of on-chip memories for the output feature map data prevents the data from being overwritten before being loaded by the ORWC unit. In other words, the ORWC unit starts to load the written output feature map data when the write operation of the output feature map data in the OW unit is completed, and then the OW unit performs a write operation on another on-chip memory, while the ORWC unit loads the written data from the first on-chip memory. This makes the hardware operate with low latency, and without any bottleneck on resource-constrained edge devices because the on-chip memory is small owing to the on-chip memory management strategy. 

## 5. Experimental Results

Qualitative and quantitative evaluations of the model presented in Section 3 were performed, which indicated that the model successfully carried out object detection regardless of the shape, number, color, and angle of the objects. Moreover, the hardware with the proposed architecture in Section 4 has been implemented in FPGA with less resource utilization and energy consumption compared to related studies, which indicates that the implemented hardware accelerator enables the presented model to be operated with low resource use and energy consumption in real time on the resource-constrained and battery-operated edge devices.

### 5.1. Performance of Proposed Model

The presented model conducted object detection on a dataset with 13,041 GT, as shown in Table 1. It manifests a recall performance of 93.1% with true positive (TP) and false negative (FN) components, and a precision of 82.6% with TP and false positive (FP) components, as described in Equations (2) and (3), respectively. In addition, an $F_{1}$ score of 87.5 is obtained according to F-measure, the metric considering both recall and precision, as described in Equation (4). In addition, the model was quantized with an unsigned integer of 8 bits for activation, integer of 8 bits for weight, integer of 32 bits for bias, and unsigned integer of 12/8 bits for the mult/shift scale, as shown in Table 2. Besides, the presented model has shown successful performance of qualitative evaluation, as shown in Figure 10, which demonstrates that the model can detect objects in diverse conditions.
(2)$$Recall=\frac{TP}{TP+FP}$$
(3)$$Precision=\frac{TP}{TP+FP}$$
(4)$$F_{1}=2*\;\frac{Precision\times Recall}{Precision+Recall}$$

### 5.2. Implementation Results and Comparison

The hardware based on the architecture proposed in Section 4 was designed using Verilog hardware description language (HDL) and verified by the coincidence of results between the C model of the presented model in Section 5.1 and register transfer level (RTL) simulation. Furthermore, it was implemented on an FPGA and verified in KU085 with the C model.

The system configuration for the experiment is illustrated in Figure 11. The camera sends pixel data to the MPSoC board, resizing the image to the size required by the model on the FPGA. In addition, the camera viewer software in the PC provides the parameter and base address to the MPSoC board and sends them to the FPGA using a serial peripheral interface (SPI). The proposed design on the FPGA starts to operate by fetching pixel data and parameters from an external memory on the FPGA board, and the design is verified by a Vivado logic analyzer utilizing an integrated logic analyzer (ILA) on the FPGA.

Table 3 presents a performance comparison with related works. The hardware resource utilization, particularly in BRAM and DSP, in [35] is quite high compared to those of [37], leading to high power consumption according to the total on-chip power in [38]. The BRAM and DSP consumed approximately 60% of the total power. In addition, the runtime is so slow that it is impossible to deploy a deep learning model on this hardware in real time owing to the performance of only 1 fps, although the logic size is also large in contrast to [37]. Consequently, the energy consumption is too high, making it unsuitable for edge devices.

To deploy deep learning hardware accelerators on edge devices, the hardware size must be small with a lightweight design owing to limited resource constraints. However, the hardware size of [36] and [38] is too large over thousands for all components, 324.7 k and 315.4 k as LUT, FF for [36], and 135.5 k as FF for [38] in VC709. The BRAM and DSP utilization is also high, so their power consumption is quite high as 27.7 W for [36] and 8.9 W for [38]. They are not able to be embedded in resource-constrained edge devices because of an enormous amount of hardware utilization and cannot be operated on battery-operated IoT devices because they require high power consumption. 

The hardware utilization for [37] in ZC702 is reasonable to be executed on low-cost devices, but the latency performance is quite poor. Therefore, it performs at only 2.62 fps, and it cannot be deployed in real time on IoT devices. Furthermore, its energy consumption is 497.6 mJ owing to its slow run time. Battery-operated devices cannot endure this hardware, which consumes high energy even though it is capable of being embedded on edge devices. 

However, the proposed hardware design has extremely low utilization in BRAM and DSP owing to a memory-efficient on-chip memory management strategy and simple but powerful 3D tensor-like PE structure with a matrix generation scheme of the input feature map. Specifically, it achieves 1.25× and 4.27× smaller logic and BRAM size, respectively, and its energy consumption is approximately 10.37× lower than the previous low-cost hardware [37] with 43.95 fps as a real-time process under an operating frequency of 100 MHz on ZC702. This indicates that the proposed hardware can be embedded in resource-constrained edge devices and deployed in the battery-operated IoT devices for object detection in real time.

By applying parallel processing between layers and involving the maxpool function in convolution operation, latency is significantly reduced, as shown in Table 4. It was experimented in RTL simulations at conv6 and conv7 in an expand layer. This demonstrates that the proposed method achieves 2.19× faster latency than the conventional method of performing convolutions in series with an additional maxpool operation. In addition, the throughput performance was improved by 2.18× as well. It indicates that low-latency and high-throughput hardware with small size can be obtained with the proposed methods. 

Table 5 presents values indicating that the proposed hardware is capable of performing increased high-speed operations with an increase in its clock frequency, resulting in a higher fps performance as well. Based on the characteristics and environment of the devices to be utilized, the proposed hardware can be executed with optimal performance on such devices.

## 6. Conclusions

With the development of deep learning technology and rapid growth in the area of IoT, deploying deep learning models on IoT devices has become an emerging field with TinyML. However, most deep learning algorithms are too complex to be executed on these resource-constrained edge devices. The algorithms are not suitable for battery-operated IoT devices owing to their high computation and energy consumption requirements. Therefore, a lightweight deep learning model and its well-optimized hardware through HW/SW co-optimization are required; hence, we proposed an optimized model for the hardware to be implemented and a lightweight and energy-efficient hardware architecture in this paper.

We presented an optimized model based on a backbone network by employing model simplification and compression. Model reduction techniques, such as involving maxpool operations in convolution, alleviates the computation complexity and latency, and hardware-oriented parameter simplification enables software and hardware to be co-optimized. Additionally, data quantization, as a model compression technique, was performed to reduce the storage space requirement for the parameters. Experiment results showed that the optimized model successfully performed object detection and was subjected to both qualitative and quantitative evaluations.

Furthermore, a lightweight and energy-efficient hardware architecture was proposed with a 3D tensor-like PE structure, generation of input feature map matrix, and a memory-efficient on-chip memory management strategy. The 3D tensor-like PE structure deals with several input feature map matrices at the same time, leading to low latency and eventually low energy consumption. In addition, logic size can be reduced owing to parallel processing between layers and channels through combination of the 3D tensor-like PE structure and input feature map matrix generation. Besides, the on-chip memory management strategy enables incremental access to an external memory without frequent and irregular data read/write operations, leading to the use of a few and small on-chip memories, and also low power consumption owing to the simple access to the memories. As a result, the proposed hardware design achieves 1.25× and 4.27× smaller logic and BRAM sizes, respectively, and consumes approximately 10.37× less energy than those of previous similar works with 43.95 fps as a real-time process under an operating frequency of 100 MHz on a Xilinx ZC702 FPGA. It indicates that the proposed hardware is capable of being embedded in the resource-constrained edge devices and can be applied to the battery-operated IoT devices.

## Figures and Tables

**Figure 1 sensors-23-01185-f001:**
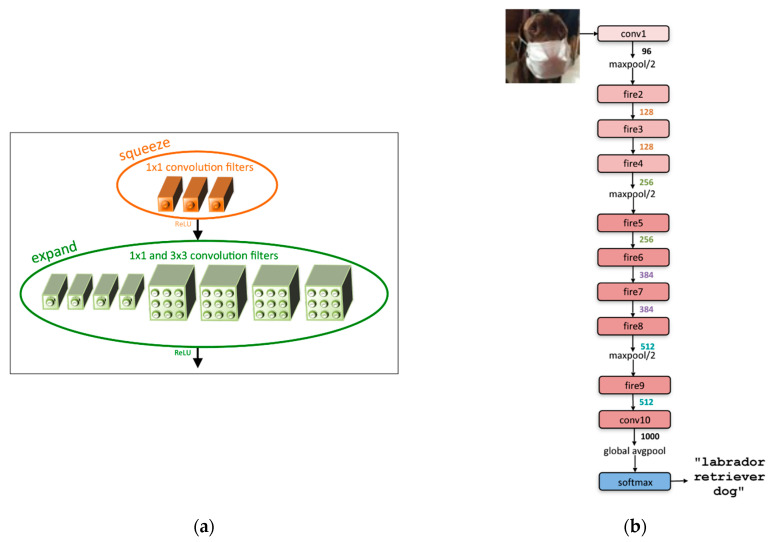
SqueezeNet architecture [21]: (**a**) micro-architectural view: fire module structure in the SqueezeNet; (**b**) macro-architectural view: SqueezeNet comprises a series of fire modules.

**Figure 2 sensors-23-01185-f002:**
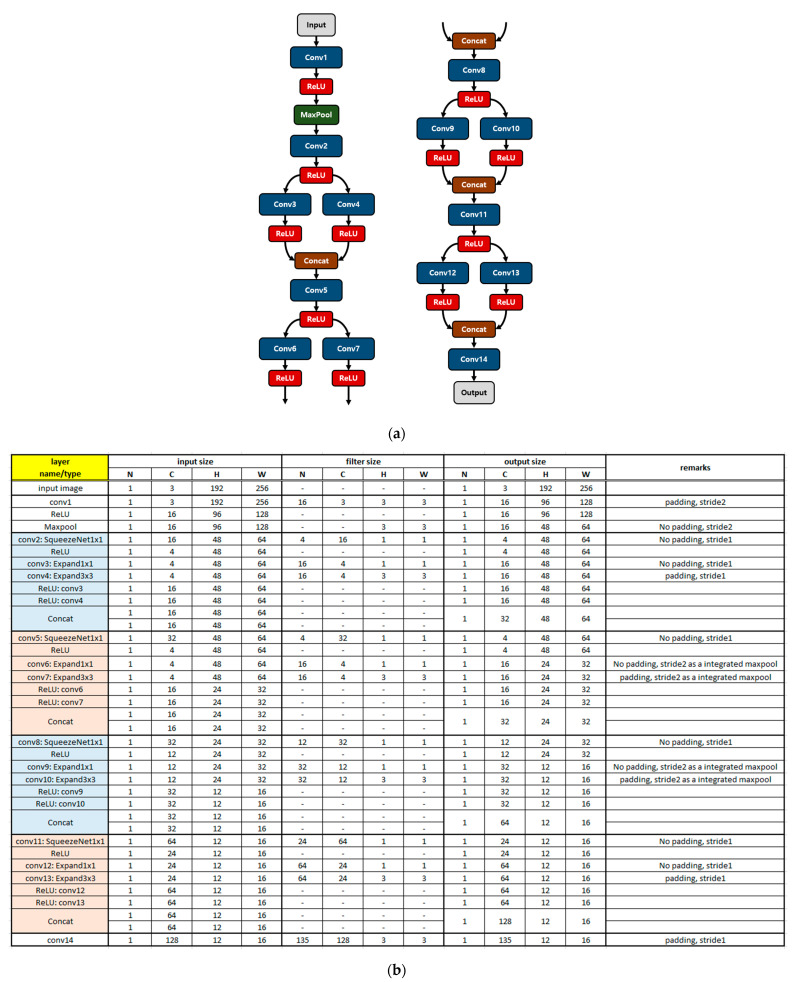
Model simplification of SqueezeNet to be more compact and smaller, being suitable for execution on the devices having limited resources: (**a**) overview of the model; (**b**) configuration of the model in detail.

**Figure 3 sensors-23-01185-f003:**
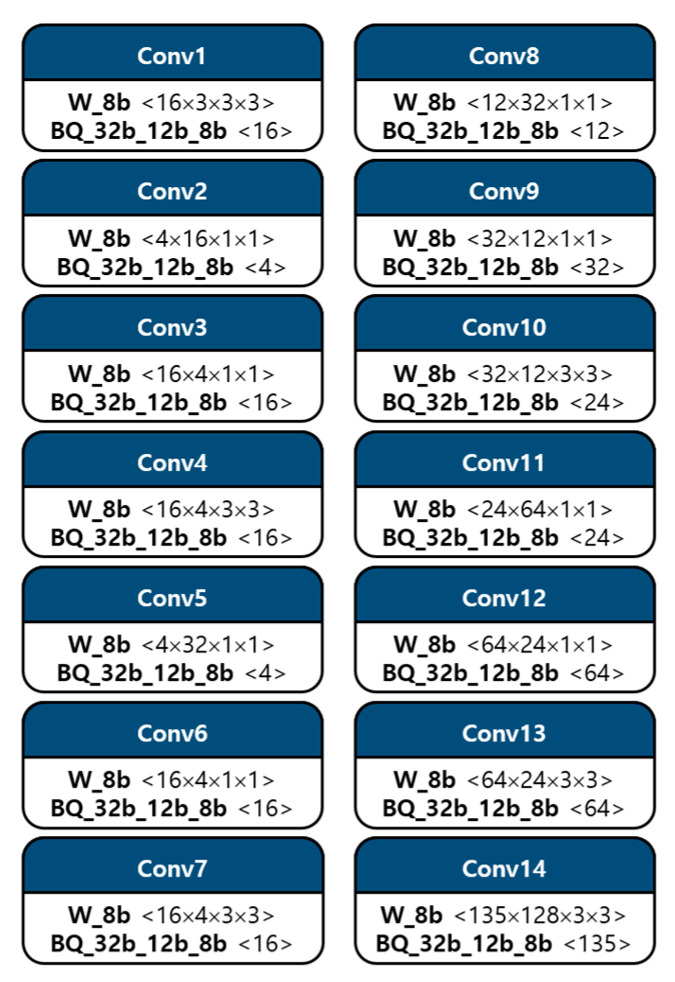
Model compression through data quantization on convolutions.

**Figure 4 sensors-23-01185-f004:**
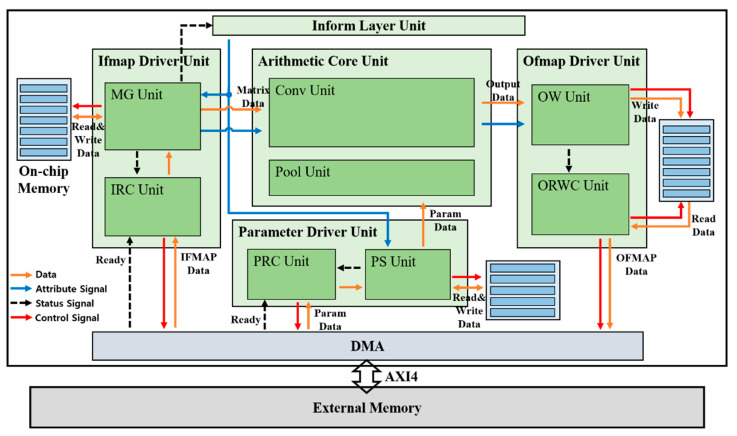
Proposed hardware architecture overview.

**Figure 5 sensors-23-01185-f005:**
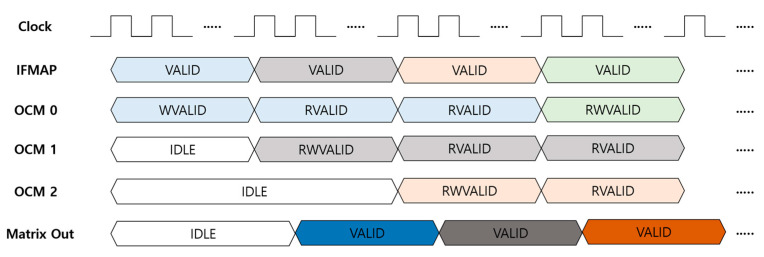
Matrix generation process in MG unit.

**Figure 6 sensors-23-01185-f006:**
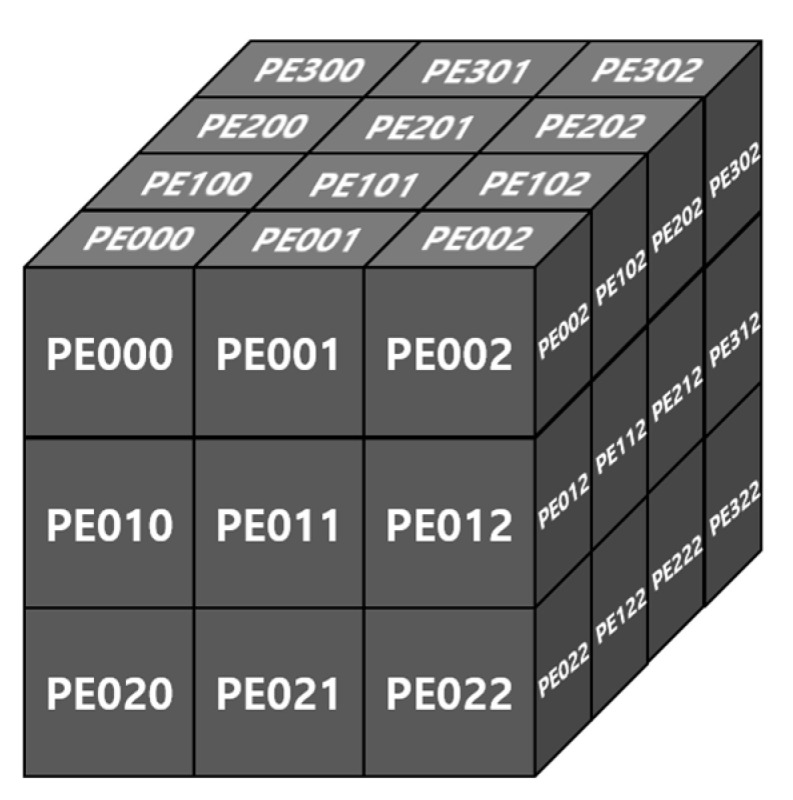
3D tensor-like PE architecture: 3 × 3 case.

**Figure 7 sensors-23-01185-f007:**
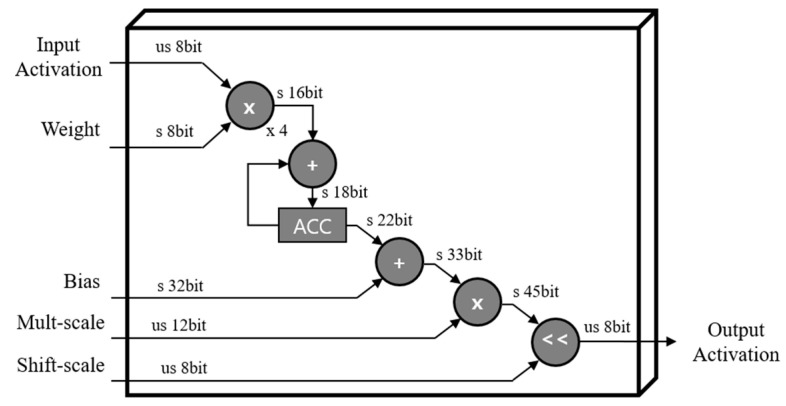
Convolution operation in PE: 1 × 1 case.

**Figure 8 sensors-23-01185-f008:**
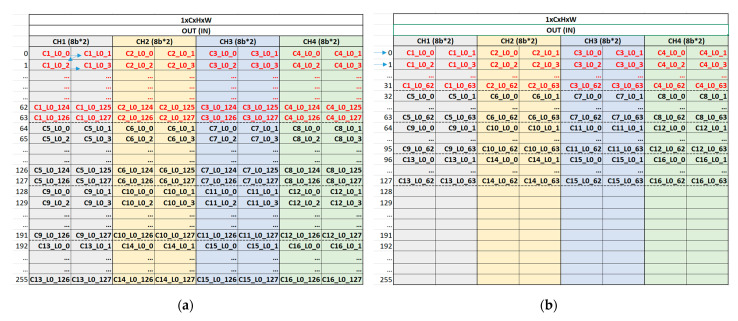
On-chip memory management strategy: (**a**) on-chip memory of OW unit at conv1 layer and its memory write operation procedure; (**b**) on-chip memory of OW unit at maxpool layer and its memory write operation procedure.

**Figure 9 sensors-23-01185-f009:**
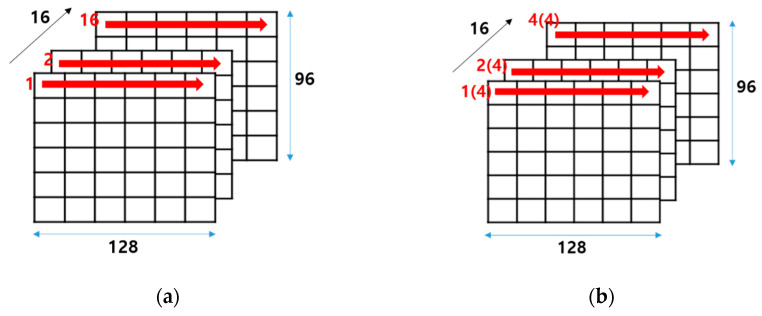
Overall process flow: (**a**) in ofmap write operation on on-chip memory at conv1 layer; (**b**) in ifmap read operation on on-chip memory at maxpool layer, and maxpool operation on maxpool unit.

**Figure 10 sensors-23-01185-f010:**
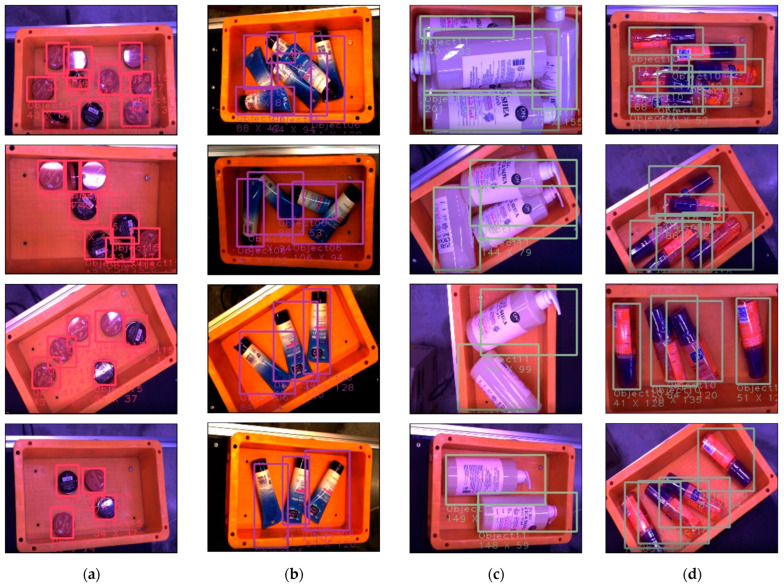
Qualitative evaluation of the performance on object detection of the proposed model. There are different objects in horizontal, and different number of objects and angle in vertical: (**a**) small objects with red and black color; (**b**) middle size of objects with blue, white, and black color; (**c**) big size of objects with white color; (**d**) middle size of objects with pink and purple color.

**Figure 11 sensors-23-01185-f011:**
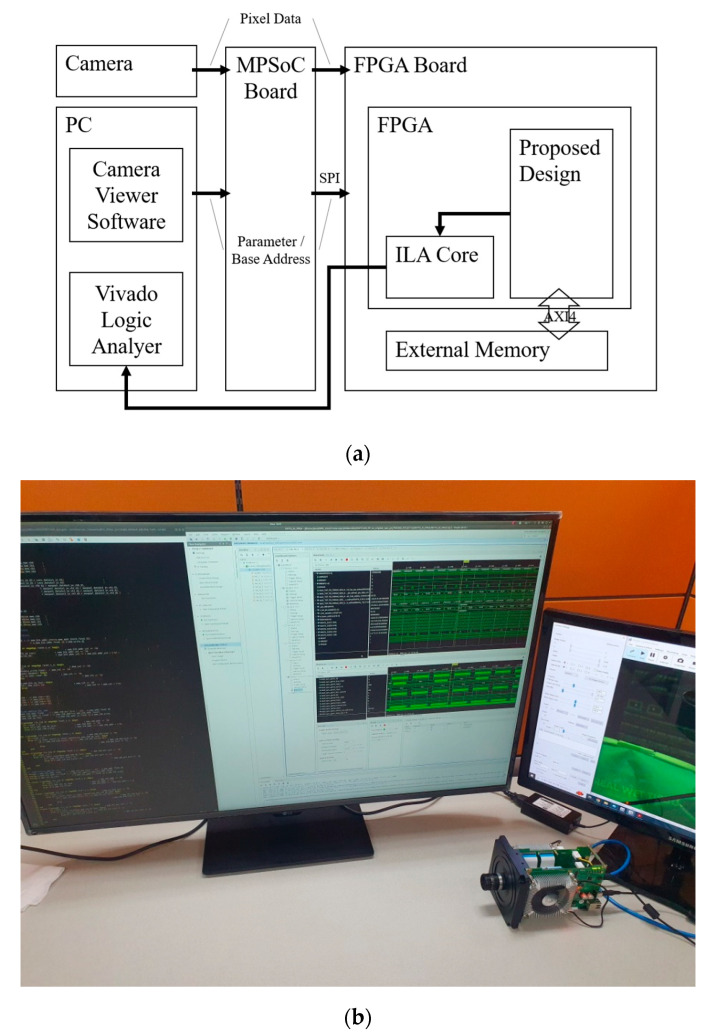
System configuration for the experiment: (**a**) block diagram; (**b**) hardware verification.

**Table 1 sensors-23-01185-t001:** Quantitative evaluation of the performance on object detection of the proposed model with an IoU threshold of 0.5 and a detection threshold of 0.697.

TP	FP	FN	GT	Recall	Precision
12,143	2561	898	13,041	93.1%	82.6%

**Table 2 sensors-23-01185-t002:** Quantization information for each component in the proposed model.

Activation	Weight	Bias	Scale
uint 8	int 8	int 32	uint 12/8 (mult/shift)

**Table 3 sensors-23-01185-t003:** Performance comparison with related works.

		[35]	[36]	[37]	[38]	Proposed ^1^
Platform		ZC702	VC709	ZC702	VC709	ZC702
Frequency (MHz)		-	110	100	100	100
Resource Utilization	LUT	54 k (102%)	324.7 k (75%)	20.2 k (37.8%)	83.6 k (19.32%)	18.3 k (34.5%)
	LUTRAM	-	-	1.2 k (7.3%)	-	5 (1%)
	FF	51 k (48%)	315.4 k (36%)	29.5 k (27.7%)	135.5 k (15.47%)	21.5 k (20.2%)
	BRAM	226 (80%)	2.7 k (92%)	134.5 (96.1%)	1.8 k (61.77%)	31.5 (23%)
	DSP	209 (95%)	1.8 k (53%)	192 (87.2%)	2.6 k (73.8%)	7 (3%)
Power (Watt)		7.95	27.7	2.23	8.9	2.11
Latency (ms)		1030	3.65	223.18	4.02	22.75
FPS (frame/sec)		1	273.97	2.62	248.76	43.95
Energy (mJ)		8188	101.11	497.6	35.78	48

^1^ Results in ZC702 for a fair comparison with [35,37].

**Table 4 sensors-23-01185-t004:** Improvement of latency and throughput performance by applying parallel processing between layers and involving maxpool layer in convolution operation.

	Conventional	Proposed
Latency (us)	271.2	124.4
Throughput (activation/us)	90.6	197.5

**Table 5 sensors-23-01185-t005:** Results of latency and fps performance according to various clock frequencies.

Frequency (MHz)	Latency (ms)	FPS (frame/sec)
167	13.62	73.4
200	11.3	88.5
220	10.15	98.5

## Data Availability

Not applicable.

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
