# Peer review of "Lightweight and Energy-Efficient Deep Learning Accelerator for Real-Time Object Detection on Edge Devices"

_sensors, 2023, doi:10.3390/s23031185_

Round 1

Reviewer 1 Report

The paper deals with a very important issue of implementing neural networks on low power devices.

The paper has a few shortcomings.

The language is sometimes difficult to understand.

You use abbreviations and acronyms that are not clearly explained in the text.

In formula 1 you left bias as a floating point value.

I really like the figure 2b - it clearly explains the structure of the network, however, contrary to what is stated in the text, the integrated maxpool operation is not clearly shown for convs 6,7,9,10. Further explanation is needed.

Figure 5 with explanation is difficult to understand. In the whole paper there are many parts that assume a priori knowledge of the issues that are not explained in the text. For example - the significance of the amount of layers to be a multiple of 4 is not explained clearlu enough.

Also the Figure 8 is really difficult to understand for someone that is not familiar with FPGA issues. Figure 9 is another example that does not mean anything even with the description provided.

In the results part you show recall and precision with true positive, false negative and such and I think the text ha to be revised here as it simply confuses the reader.

You do not provide any of those metrics for any other network that is presented in the comparison - how should we know whether your lightweight network is as capable as the other ones?

Why do you select the given numbers of bits for representation of values in the network?

How do you calculate/measure the power and energy in Table 2?

Author Response

Dear reviewer 1,

For your information, please select appropriate "Markup" option in Word file if you want to see clean text.

Reviewer 2 Report

The manuscript summarizes the background for the light-weight deep learning techniques and applications on the edge devices; and designs and implements a SqueezeNet-based simplified optimal network model to realize the light-weight and energy-efficient requirements for edge applications; the experimental results shows a much better performance than the others similar works.

The related work is important and useful, which can be applied on many of the IoT applications. The organization and presentation of the paper is also perfect, it is easy to read and understand.

Several comments or suggestions:

1, Too many keywords;

2, Please describe more clearly about the various symbols on equation (1) ?

Author Response

Dear reviewer 2,

For your information, please select appropriate "Markup" option in Word file if you want to see clean text.

We would like to thank you for your comments and recommendations on our paper. We have responded to each of your comments and recommendations as below:

Point 1: The manuscript summarizes the background for the light-weight deep learning techniques and applications on the edge devices; and designs and implements a SqueezeNet-based simplified optimal network model to realize the light-weight and energy-efficient requirements for edge applications; the experimental results shows a much better performance than the others similar works. The related work is important and useful, which can be applied on many of the IoT applications. The organization and presentation of the paper is also perfect, it is easy to read and understand.

Response 1: Thank you for your positive comment on our paper.

Point 2: Several comments or suggestions: 1, Too many keywords;

Response 2: Thank you for your comment, and a manuscript has been revised for it at Abstract.

Point 3: Several comments or suggestions: 2, Please describe more clearly about the various symbols on equation (1) ?

Response 3: Thank you for your comment, and a further clear description is stated in Section 3.2.

Reviewer 3 Report

Summary:
This paper presents an optimized network model through model simplification and compression for the hardware to be implemented,
and propose a hardware architecture for a light-weight and energy-efficient deep learning accelerator. The proposed optimized model is based on SqueezeNet [21], which is a mobile-oriented network. They perform model reduction and parameter simplification on the backbone model network through model simplification, and integer quantization is adopted for activation and parameters through model compression.  Furthermore, a light-weight and energy-efficient hardware architecture is proposed, and an implemented design is able to perform parallel processing between layers and channels deploying a 3D tensor-like processing element  structure
Strengths:
-the proposed scheme was validated using several experiments.
-the contributions are well defined in the introduction.
-the proposed hardware architecture is clearly defined
-related work section covers related architectures  
-the paper is well-written and easy to follow
Weaknesses:
-some choices of architecture are not justified.
Comments:
-it would be better if you can justify your choices of architecture, for example, in line 297 ' on-chip memories 1 and 2 are idle' why is it idle? Similarly,
-in section 5.2, it would be better if you can add F-measure in addition to recall and precision
-some related works can be added:
[1]Wang, Lin, et al. "ULO: An Underwater Light-Weight Object Detector for Edge Computing." Machines 10.8 (2022): 629.
[2]Dhelim, Sahraoui, et al. "Trust2Vec: Large-Scale IoT Trust Management System based on Signed Network Embeddings." IEEE Internet of Things Journal (2022).

Author Response

Dear reviewer 3,

For your information, please select appropriate "Markup" option in Word file if you want to see clean text.

Round 2

Reviewer 1 Report

The language still needs some corrections - please check for misspelled words. Also please correct the formatting - Table 1 is awkward to read and splits over 2 pages. Maybe it is better to enumerate the values in another, less confusing way?

In my opinion the paper presents the results of the research on an important aspect of neural networks implementation on low power devices and as such is worth publishing.

I, however, feel that some of the aspects of the chosen architecture could be better justified/justified at all.

Author Response

Dear reviewer 1,

We would like to thank you for your comments and recommendations on our paper. We have responded to each of your comments and recommendations as below:

Point 1: The language still needs some corrections - please check for misspelled words.

Response 1: Thank you for your comment, and a manuscript has been revised for it. Any revisions to the manuscript have been marked up using the “Track Changes” with the memo, ‘Round 2: It has been corrected.’.

Point 2: Also please correct the formatting - Table 1 is awkward to read and splits over 2 pages. Maybe it is better to enumerate the values in another, less confusing way?

Response 2: Thank you for your comment and recommendation, and Table 1 has been revised to make it easier to read in Section 5.1.

Point 3: In my opinion the paper presents the results of the research on an important aspect of neural networks implementation on low power devices and as such is worth publishing.

Response 3: Thank you for your positive comment on our paper.

Point 4: I, however, feel that some of the aspects of the chosen architecture could be better justified/justified at all.

Response 4: Thank you for your comment, and a further description is stated to make the chosen architecture be better justified in Section 6. Conclusions.

Reviewer 3 Report

my comments have been addressed

Author Response

Dear reviewer 3,

We would like to thank you for your comments and recommendations on our paper. We have responded to each of your comments and recommendations as below:

Point 1: my comments have been addressed

Response 1: Thank you for your comments and recommendations at Round 1. Thanks to your comments and recommendations, the paper has become richer.